# OpenReview forum: "Adversarial Tokenization"
_ICML.cc/2025/Workshop/TokShop — TokShop_

### Official Review · Reviewer_FUvJ · 2025-06-08
**AdvTok Review (Clear Accept)**

**Rating:** 8
**Confidence:** 4

**Review:**

1. Methodologies
Strengths:
Adversarial tokenization is presented as a new vulnerability in LLM safety. A multi-rooted MDD model and AdvTok algorithm are proposed. Validation is conducted across Llama3, Gemma2, and OLMo2 for jailbreaking, safety evasion, and prompt injection tasks.
Weaknesses:
Limited exploration of defense mechanisms: Only briefly discusses re-tokenization and API restrictions without testing their efficacy .
Theoretical gaps: While proving NP-hardness for tokenization optimization, the paper does not address approximation guarantees for AdvTok .
Over-reliance on greedy search: Fails to compare AdvTok’s performance against other optimization strategies
2. Data
Strengths:
Evaluated on standard adversarial benchmarks (AdvBench, Malicious, Masterkey) using StrongREJECT scores, Attack Success Rate (ASR), and safety model bypass rates.
Weaknesses:
Relies on synthetic prompts, not real-world malicious queries. Omits dataset details such as prompt length, toxicity, and biases.
3. Results
Strengths:
Adversarial Tokenization (AdvTok) shows empirical competitiveness with GCG and AutoDAN, evidenced by strong StrongREJECT scores (0.517 for Llama3 vs. 0.311). Tokenization distance correlates with attack success across models (Figure 5). Notably, ShieldGemma exhibits high bypass rates (up to 86.67%), raising safety concerns.
Weaknesses:
Limitations of Adversarial Tokenization paper:  Excludes GPT-4/Claude, limiting proprietary LLM relevance, and focuses on short-context prompts, potentially overlooking different long-context tokenization patterns.
4. Claims
Supported claims:
"Noncanonical tokenizations retain semantic signal": Validated by Q&A accuracy curves (Figure 4) .
"Adversarial tokenization bypasses alignment": Confirmed via jailbreaking success rates exceeding baselines (Table 1) .
Overreaching claims:
Paper claims "Safety pipelines are fundamentally flawed" but lacks evidence that pre-training integration fixes this. Claim of being "Competitive with SOTA attacks" is accurate for Llama3 but less so for Gemma2/OLMo2 as shown in Table 1.
Evaluation of Justification
Methods → Claims: The combination of MRMDD-based exploration and empirical validation robustly supports the core claim about tokenization vulnerabilities. However, the lack of closed-model testing weakens claims about generalizability.
Data → Results: Standard benchmarks ensure comparability but limit insights into real-world attack scenarios. Computational overhead of AdvTok has not been quantified, although not necessary, would add more weightage to the safety case at modest overhead
Results → Importance: The paper successfully argues that tokenization is a critical attack surface, justifying significance for AI safety.

---

### Official Review · Reviewer_QaoP · 2025-06-08
**Review for Adversarial Tokenization**

**Rating:** 9
**Confidence:** 4

**Review:**

**Summary**

This paper introduces AdvTok, a novel algorithm for evaluating the susceptibility of LLMs to malicious queries encoded with non-canonical tokenizations, generated using MRMDD. The authors frame the problem in terms of the edit distance between non-canonical tokenization and its canonical Byte-Pair Encoding equivalent. Through experiments on Llama3, Gemma2, and OLMo2, the authors demonstrate that query semantics are largely preserved even with non-standard tokenizations. It is interesting to see the semantic coherence decrease initially but later increase as distance increases. The authors employed the StrongREJECT evaluation framework on three adversarial tasks: jailbreaking, safety model evasion, and prompt injection. The results show that AdvTok significantly outperforms standard adversarial methods. This is further substantiated by high bypass rates against safety models like LlamaGuard and ShieldGemma. The paper concludes by hypothesizing that this vulnerability stems from pre-training. Standard safety fine-tuning uses a comparatively small dataset of canonical examples, which is insufficient to erase deeply embedded associations.

**Strengths**
1. **Well-Modeled Problem**: The problem is well-defined - the abstract concept of non-canonical tokenizations is clearly defined using MRMDD and measured using edit distance.
2. **Novel Task Introduction**: The approach of linking tokenization-level manipulations to adversarial attacks is innovative and opens new research avenues in LLM safety and robustness.
3. **Comprehensive Evaluation**: The paper is thorough, featuring extensive experiments across multiple models and adversarial tasks. The figures and tables are informative and effectively support the paper's claims.

**Weaknesses**
1. **Inconsistency in Algorithm 1**: Figure 3 shows the loop variables l = 0 and l = 1 for k=2. However, the loop in Algorithm 1 runs from 0 to k, instead of k-1.
2. **Clarity for a Broader Audience**: While the content of the paper is good, the writing is hard to understand for readers who are not specialists in this specific subfield.

---

### Official Review · Reviewer_1Y6a · 2025-06-09
**"Adversarial Tokenization" presents a novel and important line of research into LLM vulnerabilities. The strengths in novelty, empirical rigor, and clarity are significant. While there are limitations regarding practical applicability and scalability that warrant further investigation, the core findings are impactful. I lean towards Accept provided the authors can address some of the minor clarity points in the weaknesses.**

**Rating:** 8
**Confidence:** 4

**Review:**

The paper identifies and explores a novel vulnerability in Large Language Models (LLMs) related to how text is converted into tokens. It demonstrates that despite LLMs typically being trained on only one "canonical" tokenization for a given string, they still retain semantic understanding of alternative, "noncanonical" tokenizations. The core question the paper addresses is whether these alternative tokenizations can be exploited to bypass LLM safety and alignment restrictions without changing the underlying text of a harmful request.


**Strengths**

* **Original and Novel Attack Vector**: The paper introduces a novel concept by identifying and exploiting "adversarial tokenization" as a vulnerability in LLMs. Unlike traditional adversarial attacks that modify the visible text (e.g., adding suffixes or prefixes), this method achieves its effect without changing the underlying visible text of the harmful request, operating purely on the model's internal token representation. This distinction represents a previously overlooked and fresh inquiry into LLM safety research.

* **Significant insight into LLM Internal Behavior with empirical validation**: The paper empirically demonstrates that LLMs retain semantic understanding of "non canonical" alternatives which smoothly degrades as tokenizations move further from the canonical, while the comparatively smaller post-training safety alignment primarily focuses on canonical forms, creating the vulnerability.

* **Comprehensive Evaluation**: The proposed "AdvTok" attack is rigorously evaluated. Its effectiveness is demonstrated across multiple modern tokenizers (Llama3, Gemma2, OLMo2) and diverse adversarial datasets (AdvBench, Malicious, Masterkey). The authors present compelling case studies on jailbreaking, safety model evasion, and prompt injection, showcasing the broad applicability of the attack. Furthermore, thorough ablation studies investigate the impact of model size, prompt length, and hyperparameters, strengthening the paper's claims.

* **Clarity and Presentation**: The paper is very well-written and clearly structured.The problem statement, key concepts, and methodology are clearly explained, starting from the abstract and introduction. Figures (e.g., 1, 4, 5) are effective in visualizing key findings. The proposed algorithms (MRMDD and AdvTok) are detailed, with proofs and supplementary algorithms provided in the appendices, enhancing reproducibility.

**Weaknesses**

* **Limited Practical Applicability (Acknowledged)**: The authors rightly acknowledge that the current attack formulation relies on white-box access, specifically requiring model logits and the ability to input raw token sequences. This significantly limits its direct applicability to many widely used closed-source commercial LLMs, which typically offer only API-based text-in/text-out interfaces.

* **Computational Cost and Search Optimality**:
While the MRMDD and greedy search are sensible heuristics, the cost (e.g., approximately 4680 GPU hours for jailbreaking experiments alone)  may hinder broader adoption or exploration by researchers with limited resources. The paper also notes that performance is sensitive to the sampling budget, and in highly restrictive sampling scenarios (e.g., one neighbor), the method may not outperform baselines.

* **Varied Effectiveness Across Tokenizers**: The results (e.g., Table 1, Fig 5, Fig 10) indicate varied effectiveness across different tokenizers, with Gemma2 appearing more resilient. The paper could benefit from a deeper discussion on why certain tokenizers might be more or less susceptible. Are there specific properties of the tokenization scheme (e.g., vocabulary size, merging strategies) or model architecture/training that contribute to these differences?

* **Scalability to Larger Models**: The ablation study on model size shows a performance degradation as models scale up to 8B parameters. Given that many state-of-the-art models are significantly larger, it would strengthen the paper's impact to demonstrate effectiveness on larger models e.g., 13B.

* **Clarity on evidence for the main claim “noncanonical tokenizations retain the semantics”**:
Justification for the set size of 15 questions for probing semantic understanding and selection criteria would be helpful for assessing the generalizability of this finding.
From figure 4, the semantic signal sometimes increases or remains flat e.g. for Gemma2(easy and hard), but the paper lacks discussion on the potential reasons.

---

### Decision · Program_Chairs · 2025-06-10

Accept